# Extrusion and Dislocation in Titanium Middle Ear Prostheses: A Literature Review

**DOI:** 10.3390/brainsci13101476

**Published:** 2023-10-19

**Authors:** Pietro Canzi, Elena Carlotto, Luca Bruschini, Domenico Minervini, Mario Mosconi, Laura Caliogna, Ilaria Ottoboni, Cesare Chiapperini, Francesco Lazzerini, Francesca Forli, Stefano Berrettini, Marco Benazzo

**Affiliations:** 1Department of Clinical, Surgical, Diagnostic and Pediatric Sciences, University of Pavia, 27100 Pavia, Italy; 2Department of Otorhinolaryngology, University of Pavia, Fondazione IRCCS Policlinico San Matteo, Viale Camillo Golgi 19, 27100 Pavia, Italy; 3Otolaryngology, ENT Audiology and Phoniatrics Unit, University Hospital of Pisa, 56126 Pisa, Italy; 4Orthopedics and Traumatology Clinic, IRCCS Policlinico San Matteo Foundation, 27100 Pavia, Italy

**Keywords:** middle ear, ossicular replacement prosthesis, titanium, extrusion, dislocation, complication, hearing

## Abstract

Titanium middle ear (ME) prostheses are widely used in surgical practice due to their acoustic properties. However, they present a significant drawback shared by all synthetic materials currently in use for ME reconstruction: they can be rejected by the organism of the host. In this study, we aim to review the current literature on titanium partial ossicular replacement prostheses (PORPs) and total ossicular replacement prostheses (TORPs) extrusion and dislocation. Eighty articles were analysed after a full article review based on the inclusion and exclusion criteria. The most common indication for reconstruction was chronic otitis media with cholesteatoma. The average extrusion or dislocation rate was 5.2%, ranging from 0 to 35%. The average improvements in the air–bone gap were 12.1 dB (1.6 dB to 25.1 dB) and 13.8 (−0.5 dB to 22.7 dB) for the PORP and TORP groups, respectively. The data reported on this topic are highly variable, demonstrating that functional outcomes are difficult to predict in clinical practice. We believe that the current limitations could be overcome with technological developments, including bioengineering research focused on promoting prosthesis adaptation to the ME environment.

## 1. Introduction

The restoration of hearing conduction entwined within the eradication of middle ear (ME) pathologies represents an outstanding goal of modern otosurgery. Ossiculoplasty (OPL) is a surgical technique employed to restore conductive hearing loss by replacing the defective ossicular chain, of which the history dates back to the 1950s. Several materials and techniques have been adopted over the years for ossicular reconstruction, with varying degrees of success. When available, autologous material refashioned to provide customized interposition is the prime source of repair because of the high biocompatibility [1]. The most commonly used autograft material has been the reshaped incus body, although its usage is limited by the lack of availability in diseased ME, the risk of residual cholesteatoma, and the possible fixation to the walls of the ME. Until 1987, homograft implants processed from cadavers were in use and then abandoned due to the potential risk of infection [2]. Nowadays, bioengineering products from decellularized banked cortical bone are being explored [3]. In the last three decades, many alloplastic materials have been tested as suitable ossicular replacements, such as metals, ceramics, and plastics [2]. Depending on the defects to be reconstructed, synthetic prostheses are distinguished between partial ossicular replacement prostheses (PORPs), mounted on the intact stapes, and total ossicular replacement prostheses (TORPs) for cases with no stapes superstructure [4].

Since the 1990s to date, technologies that bypass the ME sound-conducting mechanism have spread within the market [5,6]; however, passive ME implants continue to be widely used and studied. At present, titanium is widely being used for its physical properties, being lightweight with high stiffness and allowing for easy individual adaptation to anatomical variations. Moreover, it is safe during magnetic resonance imaging both at 1.5 and 3 tesla, which could be necessary for the depiction of recuring cholesteatomas [7]. In current literature, numerous studies focus on the hearing outcomes with titanium prostheses, demonstrating the reliably of this material in restoring the conduction mechanism [8,9]. In a recent meta-analysis, Kortebein et al. reported a rate of 70.7% PORP and 57.1% TORP patients with a postoperative ABG less than 20 dB [8]. Otherwise, in our opinion, the rate of complications to be expected represents a notable aspect of presurgical counselling, of which the available data are controversial.

We aim to review existing literature about the complications of titanium prosthesis ossicular reconstruction, with particular regard to extrusion and dislocation, and discuss the current clinical perspectives.

## 2. Materials and Methods

A search of the literature was performed using the PUBMED database. The strategy used was based on the “Preferred Reporting Items for Systematic Reviews and Meta-analyses” (PRISMA) guidelines. The research was conducted using the following keywords: “((TORP) OR (PORP) OR (ossicular replacement) OR (ossiculoplasty)) AND titanium”. The search was conducted in June 2023. The population of interest was patients who had undergone ossiculoplasty with titanium PORP or TORP. There was no limitation in age, sex, or follow-up periods. The studies also had to include data about complications, namely prosthesis extrusion or dislocation. Only articles in English were included. Review articles, letters, editorials, and case reports were omitted. The exclusion criteria also applied to animal research and cadaver studies. In the included articles, we considered postoperative improvement in the air–bone gap (ABG) as a secondary outcome. Two reviewers performed study selection independently and screened all the retrieved titles and abstracts. The differences in assessment for analysis were resolved with assistance from a third assessor. A final selection of the articles for granular analysis was then made by the-above-mentioned two reviewers. The data extraction was performed by four authors. The extracted data were recorded in pre-developed tables in MS Excel and Word.

## 3. Results

A PRISMA flow chart illustrating both the search numbers and included studies is schematically presented in Figure 1. A total of 311 records were identified. However, after excluding articles written in languages other than English, editorials, letters to editors, case reports, and reviews, 243 studies were considered. A preliminary selection was based on titles and abstracts: 127 papers were evaluated for a full-text review, of which 42 did not meet the inclusion criteria, and 5 additional records were excluded because the full text was unavailable. Eighty articles were included in the review and were further analysed. The selected articles covered 7117 subjects and 7214 ME implants, including 3554 PORPs and 3660 TORPs. The demographic characteristics are displayed in Table 1. In six studies, the average age was less than 18 years. The most common indication for reconstruction was chronic otitis media (5829 cases, 80.8% of the total sample), with mention of cholesteatoma in 2462 cases (42.2%). The surgical details and functional outcomes are shown in Table 2. The complication rate is claimed to be 0% in 18 articles. The average extrusion or dislocation rate was 5.2%, ranging from 0 to 35%. Considering the six articles in which the average age was less than 18 years, the extrusion or dislocation rate in this limited cohort was 13.7%. The onset of complications is mentioned in only 16 articles, ranging from 2 months to 8 years. The mean follow-up time was less than or equal to 12 months in 30 studies, among which 24.8% of the complications were reported. Their management is cited in 33 articles and surgical revision is reported for 219 cases. Regarding technical details, the presence or absence of cartilage interposition between the prosthesis and the tympanic membrane is mentioned in 71 articles. When this aspect is addressed, the technique of cartilage interposition is encountered in 92.4% of cases. Prosthesis placements without cartilage protection have been described as a procedural choice for the whole sample in five works. Regarding the staging procedure, in nine articles, the OPL was staged in the whole sample, presenting a complication rate of 6.5%, whereas in 19 works, the OPL was not staged at all, with a complication rate of 2.9%. The average improvement in ABG for PORP was 12.1 dB, ranging from 1.6 dB to 25.1 dB. The average change in ABG for TORP was 13.8 (from −0.5 dB to 22.7 dB).

## 4. Discussion

OPL is a well-established surgical procedure intended for auditory rehabilitation; however, its practical use collides with some technical issues still today. An ideal OPL prosthesis should ensure hearing restoration, along with being safe and stably integrated into surrounding tissues. Titanium has proven to be a reliable material in terms of easy surgical manipulation and functional hearing results [8]. However, it is not free of postoperative complications, such as dislocation or extrusion, along with non-titanium prostheses [39]. Unfortunately, the literature regarding titanium complication rates is often conflicting. We strongly believe the fragility of evidence about this topic could hinder both the therapeutic choice and the preoperative counselling. To our knowledge, this is the first work that specifically analyses the existing data about titanium TORP and PORP extrusion and dislocation rates. In their review, Kortebein and colleagues have addressed this issue. However, they focused on articles presenting at least one audiometric outcome, with a 12-month minimum follow-up, thus limiting the available data about complications [8]. On the other hand, with the present work, we provided a comprehensive overview of this topic, considering functional auditory data in the background. We considered the rate of extrusion and dislocation for combined PORPs and TORPs because only a few articles report complications separated for the two cohorts. Our literature review has demonstrated that the reported extrusion and dislocation rates vary greatly in literature, ranging from 0 to 35%, distributed over variable follow-up time. Interestingly, 24.8% of the overall complications occurred in studies with a mean follow-up time of less than or equal to 12 months. Only a few studies reported the onset time, thus allowing for a limited discussion about this topic. The only remark which deserves to be underlined is the considerable variability of these findings, ranging from 2 months to 8 years. The evaluation focused on the six paediatric studies disclosed an increased complication rate in this cohort (13.7%) when compared to the overall results (5.2%). Unfortunately, in the studies reporting cohorts composed of both adults and children, the age of patients who underwent extrusion or dislocation is not specified, precluding a fair comparison of complication rates by age. However, we can speculate, as suggested by Michael and colleagues, that the higher frequency of eustachian tube dysfunction and air way inflammation in the pediatric population could explain the greatest incidence of complications [28]. On the other hand, the exiguity of details about the circumstances in which complications have occurred hampers a proper correlation with possible influencing factors, such as ME pathologies, surgical procedures or the staging of treatment. Regarding surgical techniques, the staging procedure could be expected to ensure a more stable ME environment and limit complications. However, our data did not reflect this trend, confirming that many variables are involved in functional results. Furthermore, many authors claim that a piece of cartilage should be placed between the head of the titanium prosthesis and the tympanic membrane in order to prevent extrusion and ameliorate acoustic transfer [41,42]. The supporters of this technique believe that the use of cartilage would enhance the adaptation of the prosthesis to the inclination of the tympanic membrane, minimize recurrence of tympanic retraction, and avoid inflammatory reaction due to direct contact with the metal. On the other hand, according to other studies, the cartilage interposition is not essential [17,50]. In our systemic review, prosthesis placement without cartilage protection have been described as procedural choice for the whole sample in five articles, declaring an extrusion rate of 0%, 6.8%, 7.5%, 10%, and 23.8%, respectively (see Table 2). These data are overwhelming to interpret, even considering that the reported extrusion or dislocation rate was 0% in 18 articles overall. It can be argued that the complete absence of complication sounds unrealistic. This number is almost certainly under-reported, or the follow-up interval was too short (less than or equal to 12 months in 10 of these studies). In addition, in our analysis, we considered the preoperative and postoperative ABG as a secondary outcome, despite the presence of these data not being among the inclusion criteria. Our findings are in agreement with those of Kortebein et al., who reported an ABG improvement of 12.1 and 16.7 dB after titanium PORP and TORP placement, respectively [8]. We disclosed a high variability in the average ABG improvement among the studies included in the present review. It should be noted that the data presented in literature are widely heterogeneous and thus difficult to compare with respect to titanium prostheses models, underlying ME diseases, surgical hands and techniques, and follow-up strategies. The information about extrusions and dislocations treatment is limited; however, we found a trend in favour of a surgical revision attempt. Unfortunately, the available data do not allow to trace any cases eventually managed with other rehabilitation strategies, such as active ME prostheses. The average ABG improvement was 12.1 dB, ranging from 1.6 dB to 25.1 dB, for the PORP group, and 13.8 dB, from −0.5 dB to 22.7 dB, for the TORP group. The functional results depend on many factors, including the status of the ME environment and eustachian tube dysfunction. According to our review, the most common indication for ossicular reconstruction is represented by chronic otitis media. Indeed, recurrence or progression of the ME pathology can affect the OPL outcomes. Furthermore, patients should be aware that despite their satisfying audiologic results, titanium PORPs and TORPs present the technical limitations shared by all passive ME prostheses. Namely, they aim to restore a normal coupling between the tympanic membrane and the inner ear, thus allowing for a non-reinforced transmission of incident sound waves [86]. In fact, they entail certain anatomical and functional prerequisites that limit their use to selected situations [4]. Currently, active ME prostheses, as well as percutaneous or transcutaneous bone conduction implants, have been designed to overcome the abovementioned issues [4,5]. Nevertheless, these solutions imply higher costs and the discomfort of an external component. All in all, current technological developments place us on the horns of a dilemma: whether to shelve the experience of titanium prostheses or strive to overcome their ME adaptability and functional limits, perhaps with bioengineering tools?

## 5. Conclusions

Here, we present a comprehensive literature review focused on titanium TORP and PORP extrusion and dislocation. We believe that the variability in the reported complication rates demonstrates that the factors involved in determining the prosthesis’ adaptability in ME are highly changeable and difficult to predict. In our opinion, these considerations should be an integral part of preoperatory counselling. Moreover, we advocate that further research that using current technology could overcome the existing limits of titanium ME PORPs and TORPs to exploit their advantage in clinical practice.

## Figures and Tables

**Figure 1 brainsci-13-01476-f001:**
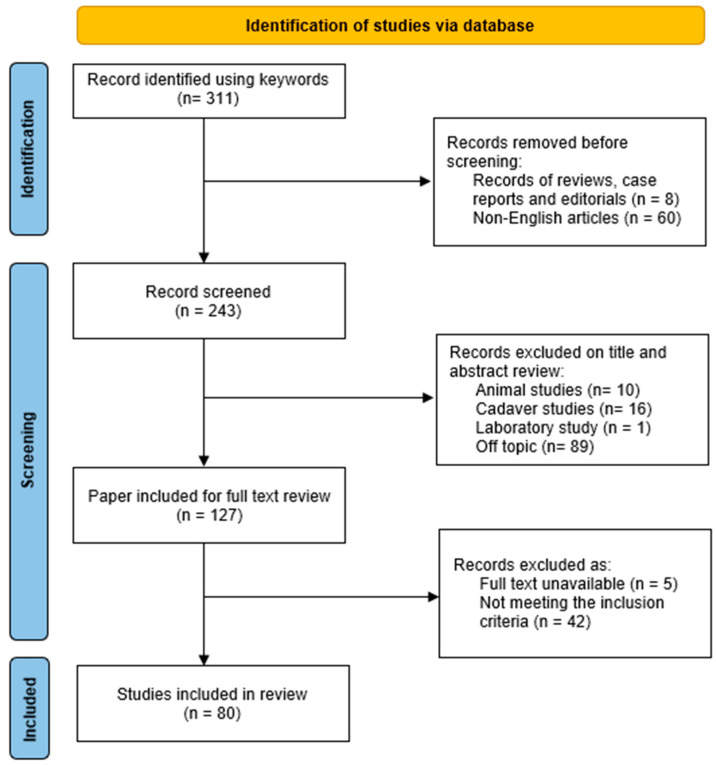
PRISMA 2020-based flow diagram showing the review search.

**Table 1 brainsci-13-01476-t001:** Demographics and clinical data.

Authors, Year	Pts	Ears	PORP	TORP	Age at Surgery (Range)	Middle Ear Disorder
Wang et al., 1999 [10]	113	124	86	38	n.a.	59 CCOM, 54 COM, 2 CA, 9 TS
Dalchow et al., 2001 [11]	1304	1304	647	657	5–82 y	1304 COM + CCOM
Krueger et al., 2002 [12]	31	31	16	15	n.a.	31 CCOM + COM
Hillman and Shelton, 2003 [13]	53	53	30	23	34.8 y	n.a.
Ho et al., 2003 [14]	25	25	14	11	45 y (14–74)	15 CCOM, 6 COM, 4 CA
Neff et al., 2003 [15]	18	18	0	18	37 y (8–80)	11 CCOM, 7 COM + TS
Neumann et al., 2003 [16]	59	61	27	34	36 y (7–81)	32 CCOM, 11 COM, 16 AT, 2 TS
Fisch et al., 2004 [17]	46	46	0	46	44.8 y (22–65)	19 CCOM, 20 COM, 7 CA + TR
Gardner et al., 2004 [18]	149	149	111	38	32.5 y (3–75)	77 CCOM, 40 COM, 32 n.a.
Martin et al., 2004 [19]	68	68	30	38	39 y (6–74)	68 CCOM + COM
Menéndez-Colino et al., 2004 [20]	23	23	0	23	37 y (8–65)	17 CCOM, 3 COM, 1 AT, 2 CA
Lehnerdt et al., 2005 [21]	15	15	14	1	12 y (n.a.)	15 CCOM + COM
De Vos et al., 2006 [22]	129	140	65	75	37 y (3–73)	72 CCOM, 12 COM, 17 AT, 16 TS, 23 n.a.
Vassbotn et al., 2006 [23]	71	73	38	35	31.5 y	40 CCOM, 4 COM. 29 CA
Schmerber et al., 2006 [24]	111	111	61	50	38.4 y (7–76)	82 CCOM, 16 COM, 4 AT, 5 CA, 1 OTO, 3 TR
Hales et al., 2007 [25]	34	34	24	10	26 y (4–59)	29 CCOM, 5 n.a.
Siddiq and Raut, 2007 [26]	33	33	20	13	35.9 y (7–64)	15 CCOM, 18 COM
Truy et al., 2007 [27]	62	62	27	35	37.5 y (n.a.)	38 CCOM, 24 n.a.
Coffey et al., 2008 [9]	80	80	29	51	28.8 y (n.a.)	35 CCOM, 36 COM, 3 CA, 4 OTO, 2 TR
Michael et al., 2008 [28]	14	14	9	5	11 y (7–14)	6 CCOM, 4 COM, 3 AT, 1 n.a.
Redaelli de Zinis, 2008 [29]	26	26	12.	14.	49 y (17–78)	26 CCOM
Alaani and Raut, 2009 [30]	97	97	65	32	39.6 y (5–75)	40 CCOM, 10 COM, 38 AT, 9 n.a.
Colletti et al., 2009 [31]	19	19	0	19	50 y (19–76)	19 COM
Roth et al., 2009 [32]	54	54	32	22	34 y (6–74)	36 CCOM, 9 COM, 4 CA, 5 TR
Woods et al., 2009 [33]	70	70	40	30	34.1 y (11–76)	33 CCOM, 12 COM, 10 AT, 15 n.a.
Fong et al., 2010 [34]	51	51	34	17	38 y (7–69)	20 CCOM, 31 COM
Inũiguez-Cuadra et al., 2010 [35]	85	94	0	94	n.a.	n.a.
Luers et al., 2010 [36]	70	70	0	70	43.6 y (6–66)	40 CCOM, 30 COM
Praetorius et al., 2010 [37]	14	14	14	0	50.4 y (13–74)	14 CCOM
Quesnel et al., 2010 [38]	74	74	27	47	11.3 y (n.a.)	n.a.
Yung and Smith, 2010 [39]	49	49	19	30	44 y (n.a.)	14 CCOM, 35 COM
Babighian and Albu, 2011 [40]	125	125	0	125	43.7 (n.a.)	125 CCOM
Beutner et al., 2011 [4]	18	18	18	0	44.4 y (8–69)	12 CCOM, 6 COM
Quaranta et al., 2011 [38]	57	57	19	38	38 y (6–67)	57 CCOM
Mardassi et al., 2011 [41]	70	70	37	33	43 y (5–77)	47 CCOM, 23 COM
Nevoux et al., 2011 [42]	114	116	0	116	9.8 y (n.a)	116 CCOM
Gostian et al., 2013 [43]	12	12	0	12	42 y (9–73)	8 CCOM, 4 COM
Hess-Erga et al., 2013 [44]	76	76	44	32	33 y (6–78)	40 CCOM, 5 COM, 31 TR
Meulemans et al., 2013 [45]	83	89	89	0	n.a. (7–85)	51 CCOM, 21 COM, 17 n.a.
Shah et al., 2013 [46]	19	19	15	4	14–50 y (n.a.)	19 CCOM + COM
Birk et al., 2014 [47]	60	62	62	0	38 y (4–83)	n.a.
Fayad et al., 2014 [48]	62	62	0	62	38.4 y (3–82)	60 COM, 2 n.a.
Órfão et al., 2014 [49]	43	43	26	17	39 y (7–70)	14 CCOM, 17 COM, 12 n.a.
Pringle et al., 2014 [50]	73	73	36	37	32.9 y (6–64)	n.a.
Baker et al., 2015 [51]	79	83	56	27	37.3 y (6–79)	74 CCOM + COM, 9 n.a.
Boleas-Aguirre et al., 2015 [52]	16	16	0	16	56 y (57–40)	8 CCOM + COM, 5 TS, 3 OTO
Lee et al., 2015 [53]	27	27	18	9	54 y (14–76)	17 CCOM, 10 COM
Ocak et al., 2015 [54]	18	18	8	10	35.2 y (13–57)	18 CCOM + COM
Roux et al., 2015 [55]	68	68	32	36	34.5 y (13–56)	68 CCOM
Wolter et al., 2015 [56]	71	75	0	75	14.3 y (7–18)	66 CCOM, 2 CA, 7 n.a.
Faramarzi et al., 2016 [57]	45	45	0	45	n.a.	10 CCOM, 3 COM, 11 TS, 21 n.a.
Gostian et al., 2016 [58]	47	47	47	0	43 y (7–73)	17 CCOM, 30 COM
Gostian et al., 2016 [59]	42	42	0	42	42.8 y (6–78)	26 CCOM, 16 COM
O’Connell et al., 2016 [60]	149	149	56	93	30.1 y (n.a.)	80 CCOM, 69 COM
Amith and Rs, 2017 [61]	20	20	20	0	25 y (12–52)	10 CCOM, 10 COM
Govil et al., 2017 [62]	101	101	47	54	9.8 y (3.4–17.3)	n.a.
McMullen et al., 2017 [63]	71	71	23	48	26 y (6–73)	n.a.
Mulazimoglu et al., 2017 [64]	126	126	126	0	37 y (7–72)	86 CCOM, 11 COM, 11 TR, 3 TS, 2 TU, 13 n.a.
Choi and Shin, 2018 [65]	45	45	20	25	n.a.	45 CCOM + COM
Kahue et al., 2018 [66]	130	130	130	0	36 y (n.a.)	121 COM, 9 TR
Kong et al., 2017 [67]	20	20	9	11	49 y (n.a.)	n.a.
Kumar et al., 2017 [68]	37	37	31	6	31.6 y (13–48)	37 CCOM + COM
Lahlou et al., 2018 [69]	256	280	163	117	44 y (17–74)	125 CCOM, 85 COM, 40 AT, 4 CA, 11 OTO, 12 TR, 3 TU
Saliba et al., 2018 [70]	158	158	0	158	29.7 y (n.a.)	103 CCOM, 25 COM, 19 CA, 11 n.a.
Gu and Chi, 2019 [71]	206	206	134	72	46 y (12–70)	206 CCOM + COM
Haidar et al., 2019 [72]	129	133	88	45	33 y (7–74)	34 COM, 10 AT, 24 GR, 65 n.a.
Potsangbam and Akoijam, 2019 [1]	20	20	14	6	30 y (8–64)	20 CCOM
Wood et al., 2019 [73]	153	153	0	153	40 y (6–89)	120 CCOM, 10 COM, 23 n.a.
Mantsopoulos et al., 2021 [74]	274	274	274	0	38 y (6–67)	168 CCOM, 62 COM, 37 TS, 6 TR, 1 TU
Baazil et al., 2022 [75]	106	106	0	106	35 y (6.6–75.3)	105 CCOM + COM, 1 CA
Bahmad and Perdigão, 2022 [76]	13	13	0	13	44 y (n.a.)	7 CCOM, 6 COM
Kraus et al., 2022 [77]	36	38	38	0	40.4 y (6–81)	18 CCOM, 20 COM
Park et al., 2022 [78]	135	135	86	49	n.a.	94 CCOM, 41 COM
Plichta et al., 2022 [79]	24	24	12	12	38.33 y (4–62)	24 TR
Van Hoolst et al., 2022 [80]	99	113	0	113	n.a. (8–87)	74 CCOM, 15 COM, 24 CA
Chien et al., 2023 [81]	8	8	8	0	n.a. (27–48)	8 TR
Faramarzi et al., 2023 [82]	248	248	115	133	33 y (n.a.)	248 CCOM + COM + TS
Gülşen and Çıkrıkcı, 2023 [83]	21	21	0	21	n.a. (28–44)	21 TS
Kálmán et al., 2023 [84]	130	130	84	46	n.a.	130 CCOM + COM
Kim et al., 2023 [85]	130	130	78	52	49.2 y (n.a.)	71 CCOM, 59 COM

Number of patients (pts), when not specified, it is assumed to be the same number of ears; years (y); not available (n.a.); chronic otitis media with cholesteatoma (CCOM); chronic otitis media without cholesteatoma (COM); atelectasis (AT); congenital abnormalities (CA); otosclerosis (OTO); traumatic injuries (TR); tympanosclerosis (TS); tumours (TU).

**Table 2 brainsci-13-01476-t002:** Surgical details and functional outcomes.

Authors, Year	Surgery	Cartilage	Staging	Pre-op ABG		Post-op ABG		Extrusions and Dislocations/Ears (%)	Onset	Treatment	Follow-Up
				PORP	TORP	PORP	TORP				
Wang et al., 1999 [10]	11 CWD, 48 CWU, 65 OPL	124	0	n.a.	n.a.	n.a.	n.a.	2/124 (1.6%)	n.a.	n.a.	12–46 mo
Dalchow et al., 2001 [11]	n.a.	1304	1100	n.a.	n.a.	14	15	29/1304 (2.2%)	n.a.	n.a.	6–72 mo
Krueger et al., 2002 [12]	n.a.	31	n.a.	n.a.	n.a.	14.1	n.a.	0/31 (0%)	/	/	16 mo–2 y
Hillman and Shelton, 2003 [13]	n.a.	53	n.a.	n.a.	n.a.	n.a.	n.a.	2/53 (3.8%)	12 mo	surgical	3 mo–n.a.
Ho et al., 2003 [14]	9 CDW, 15 CWU, 1 OPL	19	20	38.7	42.8	18.1	21.5	1/25 (4%)	n.a.	surgical	6 mo
Neff et al., 2003 [15]	16 OPL, 2 n.a.	18	6	n.a.	33.9	n.a.	16.9	0/18 (0%)	/	/	8 mo
Neumann et al., 2003 [16]	n.a.	61	0	n.a.	n.a.	n.a.	n.a.	0/61 (0%)	/	/	38 mo
Fisch et al., 2004 [17]	21 CWD, 25 CWU	0	46	/	41.9	/	20.7	0/46 (0%)	/	/	1–7 y
Gardner et al., 2004 [18]	13 CWD,68 CWU, 21 OPL, 47 n.a.	149	13	26	40	18.8	21.7	1/149 (0.7%)	n.a.	n.a.	1.5 y
Martin et al., 2004 [19]	14 CWD, 7 CWU, 47 OPL	68	4	34	34	17	25	1/68 (1.5%)	n.a.	surgical	3 mo–2.5 y
Menéndez-Colino et al., 2004 [20]	16 CWD, 7 CWU	23	0	/	n.a.	/	n.a.	0/23 (0%)	/	/	18 mo
Lehnerdt et al., 2005 [21]	n.a.	15	0	n.a.	n.a.	n.a.	n.a.	1/15 (6.6%)	n.a.	n.a.	6 mo
De Vos et al., 2006 [22]	8 CWD, 65 CWU, 67 OPL	140	24	32.7	42.5	18.1	19.8	11/140 (7.9%)	22 mo	surgical	11.8 mo
Vassbotn et al., 2006 [23]	20 CWD, 53 CWU	73	0	28	38	9	19	4/73 (5.5%)	n.a.	2 surgical, 2 n.a.	14.2 mo
Schmerber et al., 2006 [24]	18 CWD, 93 CWU	92	65	n.a.	n.a.	15	26.5	14/111 (12.6%)	1: 17 mo, 1: 20 mo, 12 n.a.	13 surgical, 1 refused	20.5 mo
Hales et al., 2007 [25]	4 CWD, 30 CWU	34	34	n.a.	n.a.	n.a.	n.a.	2/34 (5.9%)	n.a.	n.a.	19 mo
Siddiq and Raut, 2007 [26]	5 CWD, 14 CWU, 14 n.a.	33	n.a.	25.1	23.3	15.3	13.4	0/33 (0%)	/	/	6–18 mo
Truy et al., 2007 [27]	n.a.	62	n.a.	30.9	28	19.4	18.3	2/62 (3.2%)	2 mo	surgical	12 mo
Coffey et al., 2008 [9]	22 CDW, 22 CWU, 30 OPL	80	34	n.a.	n.a.	14.3	16.0	3/80 (3.8%)	n.a.	n.a.	14.9 mo
Michael et al., 2008 [28]	7 CWD, 3 CWU, 4 OPL	14	7	n.a.	n.a.	n.a.	n.a.	0/14 (0%)	/	/	12 mo
Redaelli de Zinis, 2008 [29]	26 CWD	26	0	n.a.	n.a.	n.a.	n.a.	0/26 (0%)	/	/	12 mo
Alaani and Raut, 2009 [30]	57 CWD, 40 CWU + OPL	97	29	26.3	32.1	10.6	14.8	2/97 (2.1%)	1 y	1 surgical, 1 n.a.	1 y
Colletti et al., 2009 [31]	19 CWU	19	n.a.	/	40.7	/	21.5	3/19 (15.8%)	n.a.	2 surgical, 1 n.a.	36 mo
Roth et al., 2009 [32]	11 CWD, 27 CWU, 16 OPL	54	29	31.0	38.2	13.3	16.4	1/54 (1.8%)	/	surgical	1–5 y
Woods et al., 2009 [33]	n.a.	70	3	32.2	39.2	26.9	29.2	11/70 (15.7%)	n.a.	n.a.	6 mo
Fong et al., 2010 [34]	4 CWD, 47 CWU	51	n.a.	n.a.	n.a.	n.a.	n.a.	1/51 (2%)	n.a.	n.a.	12 mo
Inũiguez-Cuadra et al., 2010 [35]	56 CWD, 38 CWU	94	n.a.	/	23.8	/	15.4	8/94 (8.5%)	n.a.	n.a.	38 mo
Luers et al., 2010 [36]	33 CWD, 37 CWU	70	0	/	33.9	/	19.7	0/70 (0%)	/	/	2 mo
Praetorius et al., 2010 [37]	n.a.	14	n.a.	26.6	/	15.2	/	0/14 (0%)	/	/	8 mo
Quesnel et al., 2010 [38]	n.a.	74	49	30.2	36.6	20.8	22	7/74 (9.5%)	n.a.	n.a.	33 mo
Yung and Smith, 2010 [39]	n.a.	49	7	29.2	32.5	16.2	20.7	8/49 (16.3%)	n.a.	n.a.	2 y
Babighian and Albu, 2011 [40]	125 CWD	125	125	/	44.9	/	22.3	2/125 (1.6%)	n.a.	n.a.	12 mo
Beutner et al., 2011 [4]	11 CWD, 7 CWU	n.a.	0	26.8	/	18.2	/	0/18 (0%)	/	/	6 mo
Quaranta et al., 2011 [38]	57 CWU	0	57	36.7	45.3	24.1	27.2	6/57 (10.5%)	n.a.	surgical	24 mo
Mardassi et al., 2011 [41]	70 CWU + OPL	70	24	27.2	32.8	15	19.5	4/70 (5.7%)	n.a.	surgical	9.8 mo
Nevoux et al., 2011 [42]	116 CWU	116	116	/	41	/	22.4	17/116 (14.7%)	2:1 y, 6:2 y, 5:5 y, 1:>5 y, 3:n.a.	surgical	34 mo
Gostian et al., 2013 [43]	1 CWD, 11 CWU	12	n.a.	/	26.6	/	18.8	0/12 (0%)	/	/	32 mo
Hess-Erga et al., 2013 [44]	31 OPL, 40 n.a.	76	n.a.	28	37	15	20	4/76 (5%)	<1 y	2 surgical, 2 conservative	5.2 y
Meulemans et al., 2013 [45]	7 CWD, 65 CWU, 17 OPL	89	0	26.2	/	15.6	/	3/89 (3.4%)	/	surgical	13 mo
Shah et al., 2013 [46]	19 CWU	19	0	n.a.	n.a.	n.a.	n.a.	0/19 (0%)	/	/	11.1 y
Birk et al., 2014 [47]	n.a.	n.a.	n.a.	26.9	/	15.4	/	1/62 (1.6%)	n.a.	n.a.	7 mo
Fayad et al., 2014 [48]	8 CWD, 21 CWU, 30 OPL, 3 n.a.	62	23	/	35.1	/	18	1/62 (1.6%)	1 y	n.a.	21.7 mo
Órfão et al., 2014 [49]	1 CWD, 11 CWU, 31 OPL	43	0	32.8	37.1	21.9	25.7	1/43 (2%)	n.a.	n.a.	20 mo
Pringle et al., 2014 [50]	9 CWD, 47 CWU, 17 OPL	0	52	n.a.	n.a.	n.a.	n.a.	5/73 (6.8%)	n.a.	surgical	6–96 mo
Baker et al., 2015 [51]	n.a.	83	38	28.2	30.3	16.5	20.6	5/83 (6.0%)	n.a.	surgical	41.8 mo
Boleas-Aguirre et al., 2015 [52]	2 CWD, 2 CWU, 12 OPL	16	n.a.	/	34	/	16.4	0/16 (0%)	/	/	12 mo
Lee et al., 2015 [53]	15 CWD, 11 CWU, 1 OPL	n.a.	n.a.	23	28	12	15	0/27 (0%)	/	/	6 mo
Ocak et al., 2015 [54]	15 CWU, 3 OPL	16	n.a.	33.7	38	8.6	19	1/18 (5.5%)	12 mo	n.a.	38.5 mo
Roux et al., 2015 [55]	68 CWU	68	19	23.5	31	19.5	26	0/68 (0%)	/	/	23 mo
Wolter et al., 2015 [56]	n.a.	75	n.a.	/	44	/	30	1/75 (1.3%)	3.9 y	surgical	2.7 y
Faramarzi et al., 2016 [57]	45 OPL	n.a.	45	/	36	/	24.7	2/45 (4.4%)	n.a.	n.a.	6–12 mo
Gostian et al., 2016 [58]	15 CWD, 32 CWU	47	n.a.	25.7	n.a.	16.8	n.a.	3/47 (6.4%)	n.a.	1 n.a., 2 surgery	6.5 y
Gostian et al., 2016 [59]	18 CWD, 24 CWU	42	0	/	33	/	22	3/42 (7.1%)	6 mo	surgical	6.8 y
O’Connell et al., 2016 [60]	n.a.	149	77	30.9	37.9	17.6	21.8	5/149 (3.2%)	n.a.	surgical	51.6 mo
Amith and Rs, 2017 [61]	3 OPL, 17 n.a.	20	3	44.4	/	31.3	/	3/20 (15%)	n.a.	n.a.	12 mo
Govil et al., 2017 [62]	n.a.	89	n.a.	n.a.	n.a.	n.a.	n.a.	29/101 (29%)	1.3 y	surgical	2.2 y
McMullen et al., 2017 [63]	n.a.	71	50	n.a.	n.a.	n.a.	n.a.	2/71 (2.8%)	8 mo, n.a.	1 conservative, 1 surgical	10.2 mo
Mulazimoglu et al., 2017 [64]	33 OPL, 93 n.a.	0	33	/	28.2	/	22.3	30/126 (23.8%)	26 mo	12 surgical, 18 n.a.	4.5 y
Choi and Shin, 2018 [65]	n.a.	45	8	n.a.	n.a.	n.a.	n.a.	2/45 (4.4%)	/	/	12 mo
Kahue et al., 2018 [66]	5 CWD, 77 CWU, 48 OPL	130	10	29	/	18	/	3/130 (2.3%)	25 mo	surgical	37 mo
Kong et al., 2017 [67]	n.a.	20	n.a.	27.6	28.5	26	29	4/20 (20%)	n.a.	n.a.	27 mo
Kumar et al., 2017 [68]	n.a.	37	0	32.1	37.5	26.3	21.6	1/37 (2.7%)	n.a.	n.a.	6 mo
Lahlou et al., 2018 [69]	46 CWD, 74 CWU, 160 OPL	280	0	26	30	15	20	17/280 (6.1%)	n.a.	surgical	12 mo
Saliba et al., 2018 [70]	36 CWD, 122 CWU	158	62	/	38	/	26.3	24/158 (15%)	n.a.	surgical	18 mo
Gu and Chi, 2019 [71]	206 CWD	n.a.	0	27.8	28	16.4	18.4	2/206 (1.0%)	/	/	30 mo
Haidar et al., 2019 [72]	133 CWU	133	8	29.6	33.3	12.9	18.7	5/133 (2.3%)	n.a.	n.a.	6 mo
Potsangbam and Akoijam, 2019 [1]	20 OPL	20	20	37.1	40.3	17.6	22.5	7/20 (35%)	4–8 y	n.a.	3 y
Wood et al., 2019 [73]	24 CWD, 114 CWU, 15 OPL	152	14	/	37.5	/	24.9	2/153 (1.3%)	n.a.	surgical	44 mo
Mantsopoulos et al., 2021 [74]	n.a.	n.a.	n.a.	22.7	/	15.7	/	8/274 (2.9%)	n.a.	n.a.	4 mo
Baazil et al., 2022 [75]	3 CWD, 33 CWU, 70 OPL	0	74	/	38.1	/	27.6	8/106 (7.5%)	n.a.	n.a.	11.7 mo
Bahmad and Perdigão, 2022 [76]	13 CWD	n.a.	n.a.	/	35.1	/	20.7	0/13 (0%)	/	/	6 mo
Kraus et al., 2022 [77]	n.a.	7	0	21.8	/	10.5	/	2/38 (5.3%)	n.a.	n.a.	1–9 y
Park et al., 2022 [78]	113 CWD, 22 CWU	135	0	28.1	28.1	18.4	23.5	1/135 (0.7%)	n.a.	n.a.	8.1 mo
Plichta et al., 2022 [79]	24 OPL	24	n.a.	n.a.	n.a.	n.a.	n.a.	1/24 (4.1%)	n.a.	surgical	24 mo
Van Hoolst et al., 2022 [80]	28 CWD, 23 OPL, 62 n.a.	113	n.a.	/	32.7	/	21.7	17/113 (15.0%)	n.a.	surgical	39 mo
Chien et al., 2023 [81]	8 OPL	n.a.	0	25.9	/	10.8	/	0/8 (0%)	/	/	3.58 mo
Faramarzi et al., 2023 [82]	248 OPL	n.a.	248	34	37.6	21.2	24.6	7/248 (2.8%)	n.a.	n.a.	12.5 mo
Gülşen and Çıkrıkcı, 2023 [83]	21 OPL	21	0	/	37.1	/	14.5	1/21 (4.8%)	n.a.	surgical	12 mo
Kálmán et al., 2023 [84]	n.a.	130	n.a.	n.a.	n.a.	n.a.	n.a.	11/130 (8.5%)	n.a.	surgical	8.8 mo
Kim et al., 2023 [85]	130 OPL	130	130	n.a.	n.a.	n.a.	n.a.	10/130 (7.7%)	n.a.	n.a.	22.7 mo

Canal wall down (CWD); canal wall up (CWU); ossiculoplasty (OPL); years (y); months (mo); not available (n.a.). Data included in the column “surgery” refer to the procedure performed at the time of the ME implant positioning, regardless of any previous surgery. In the column “cartilage”, we included the case in which cartilage interposition between the prosthesis and the tympanic membrane was cited. In the column “staging”, we reported, when available, the number of ossicular reconstruction performed in a second-stage procedure. In the columns “preoperative ABG (pre-op ABG)” and “postoperative ABG (post-op ABG)”, n.a. indicates that this outcome is not mentioned or audiometric results differentiated between the PORP and TORP groups are not available. “Extrusions and dislocations/ears (%)”: we excluded extrusions and dislocations caused by cholesteatomas. The data included in the columns “Onset” and “Treatment” refer to the time of appearance and management of complications. “Follow-up” indicates the average follow-up or, when not available, its range.

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
