# Peer review of "Extrusion and Dislocation in Titanium Middle Ear Prostheses: A Literature Review"

_brainsci, 2023, doi:10.3390/brainsci13101476_

Round 1

Reviewer 1 Report

This manuscript is puzzling and is suffering of many flaws. It is really an important review on the use of titanium ossicle prosthesis to restore hearing mainly in chronic middle ear otitis. It concerns 81 articles published from 1999 to 2023, although 2 systematic reviews were issued this year , Kortebein et al. (ref. 7) and Omar et al.(Am J Otolaryngology 2023 Jan-Feb, not quoted), which reviewed 40 and 11(children) articles, respectively. In both  reviews, the audiometric data were accurately provided, far better than in the present work which are limited to the mean residual air-bone gap and minimal and maximal values. No other audiometric information are given herein.

In the introduction and discussion, the reference to the relationship between cognitive ressources and middle ear reconstruction is inappropriate as no study has been undertaken up to now on this topic. Indeed it concerns hearing restoration for sensorineural hearing loss, specially by cochlear implantation as quoted in ref 1,86,87. As well restoration of hearing in rare chronic middle ear otitis by either cochlear or middle ear implant is not the subject of this study (ref 3,4,6). At variance, an historical review on ossiculoplasty since the remodeled patient or sampled cadaver ossicles to the titanium ones would better introduce the present study. For each previous ossicle types, the rate of extrusion or dislocation would be of importance to compare the benefit achieved, if any, by using titanium prosthesis during the last 2 decades. To what extent covering any prosthesis by cartilage has been the reason of less extrusion ? It would be interesting to indicate how the titanium prosthesis was protected from the eardrum. This should be easy to find in the 81 articles and to compare the extrusion/dislocation rates between those prosthesis which have been protected eand the others not.

In the methods, the authors have excluded the cases of prosthesis exclusion due to cholesteatoma recurrence. This is questionable as most of these recurrences are in relation of eardrum reconstruction (insufficient cartilage recovering of the middle ear cleft and/or persistence of Eustachian tube dysfunction) rather than a remnant of cholesteatoma in the middle ear cavities. These cases should be analysed in this study.

In the results, why the residual ABG appears first (line 85) which is a second objective of the study, and then the extrusion/dislocation rate (lines 86-89)?

Minor comments:

line 17: TORO AND PORP to be explained like in lines 48-49.

line 67: not to be excluded, see above.

line 70: audiometric data to be given in details, see above.

line73: were instead of "ware".

lines 82-84: it would be more informative to indicate that the presence of cholesteatoma was identified in 2/3 of 60,3% of chronic otitis media, although it was not possible in the 2300 others cases.

Lines 86-87: it would be interesting to indicate whether the extrusion are similar in adults and children.

Table 1: CHL and TU not explained in the figure legend.

Lines 168-169: rather than to consider "a foreign body inflammation" the importance of a good cartilage covering of the prothesis should be discussed to avoid both an atelectasis recurrence and a prosthesis extrusion/dislocation.

No problem.

Author Response

We thank the Reviewer for the time devoted to our research and for his suggestions. In the following we address all the issues raised, highlighting the modification to the manuscript.

1-2 systematic reviews were issued this year, Kortebein et al. (ref. 7) and Omar et al. (Am J Otolaryngology 2023 Jan-Feb, not quoted), which reviewed 40 and 11(children) articles, respectively. In both reviews, the audiometric data were accurately provided, far better than in the present work which are limited to the mean residual air-bone gap and minimal and maximal values. No other audiometric information are given herein.

Thank you for your comment. Our work aims to review the existing literature about complications of titanium prostheses ossicular reconstruction, using as inclusion criteria the mention of extrusion and dislocation. In this respect, we have included the audiometric data as an essential functional aspect but we considered it as a corollary. On the other hand, Kortebein et al. used as inclusion the articles presenting at least one audiometric outcome, with a 12-months minimum follow up, thus excluding some work which do not reported auditory data but mentioning extrusion or dislocation. We have stressed this point in the discussion following your suggestion. Notably, in our systematic review 24.8% of the overall complications occurred in studies with a mean follow-up time of less than or equal to 12 months. Conversely, we have not cited the article of Omar et al. because it deals with ossicular prostheses regardless the constituent material, among the 11 articles analysed just one (Quesnel et al 2010, included in our review) regards titanium prostheses.

2- In the introduction and discussion, the reference to the relationship between cognitive ressources and middle ear reconstruction is inappropriate…

We have revised this section as suggested.

3- It would be interesting to indicate how the titanium prosthesis was protected from the eardrum. This should be easy to find in the 81 articles and to compare the extrusion/dislocation rates between those prosthesis which have been protected and the others not.

Thank you for your suggestions: we added the table 2 a specific column and the topic was discussed according to your comments.

4- In the methods, the authors have excluded the cases of prosthesis exclusion due to cholesteatoma recurrence..

This aspect was revised as suggested.

5- In the results, why the residual ABG appears first (line 85) which is a second objective of the study, and then the extrusion/dislocation rate (lines 86-89)?

Thank you very much for highlighting this aspect. We have modified the results as suggested.

6- line 17: TORP AND PORP to be explained like in lines 48-49.

Thank you, we have explained it.

7- line 67: not to be excluded, see above.

Thank you for the comment, we have added these cases.

8- line 70: audiometric data to be given in details, see above.

Thank you for your comment. Our work aims to review the existing literature about complications of titanium prostheses ossicular reconstruction, using as inclusion criteria the mention of extrusion and dislocation. In this respect, we have included the audiometric data as an essential functional aspect but we considered it as a corollary.

9- line 73: were instead of "ware".

Thank you.

10- lines 82-84: it would be more informative to indicate that the presence of cholesteatoma was identified in 2/3 of 60,3% of chronic otitis media, although it was not possible in the 2300 others cases.

Thank you we have modified these sentences.

11- Lines 86-87: it would be interesting to indicate whether the extrusion are similar in adults and children.

The data provided in current literature allow to measure the extrusion or dislocation rate in the limited cohort of the six articles in which the average age was less than 18 years (see line 84). Indeed, we have made a comparison limited to this works. Unfortunately, the studies involving cohorts composed of both adult and children did not specify the age of patients who underwent extrusion or dislocation. As result, a fair comparison of complication rates by age is impossible.

12- Table 1: CHL and TU not explained in the figure legend.

Thank you, we have provided corrections.

Lines 168-169: rather than to consider "a foreign body inflammation" the importance of a good cartilage covering of the prothesis should be discussed to avoid both an atelectasis recurrence and a prosthesis extrusion/dislocation.

Thank you for identifying these additional aspects to strengthen the work. We added to discussion a few remarks concerning your suggestions.

Reviewer 2 Report

This study reviewed the literature on ossiculoplasty with a focus on titanium TORP and PORP extrusion and dislocation and postoperative improvement of the air-bone gap. Systematic review articles should follow general principles. Although the Methods section of this article has been described according to the Preferred Reporting Items for Systematic Reviews and Meta-analyses (PRISMA) guidelines, the overall presentation is unsatisfactory. It is strongly recommended that authors refer to several systematic reviews published in top journals in their field and use them as templates for format and content to reorganize their articles. Here are some specific suggestions.

1.       Firstly, the Introduction section consisted of a single paragraph, which was relatively lengthy and made it easy to lose sight of the main points. The Introduction section of a systematic review article usually has a few catchy paragraphs that are structured and briefly describe the topic or issue surrounding the article.

2.       The Methods section did not provide a clear description of the steps and content of the review, such as the date of the last search, the role of independent reviewers in the selection of articles, the software tools used, and the assessment of the quality and bias of the included articles. In addition, the search of the PubMed database alone was incomplete, with only 311 papers found in the initial search as described in the article.

3.       Results section: "average extrusion or dislocation rate was 5.1%", "average improvement in ABG for PORP was 12.3 dB", and "average change in ABG for TORP was 13.7", were these average rates or values calculated by performing a meta-analysis? If so, a detailed description of the statistical model, heterogeneity, and software tools would be required in the Methods section, as well as presented as a forest plot in the Results section.

4.       Minor:

a) The text, such as "(-0,5 dB to 22.7 dB)" and "(CCOM, 34,1%)", appeared in several places, which should be -0.5 and 34.1%;

b) The format of references such as "[3], [4]" (Line 41) and "[6], [3]" (Line 179) appeared in two places in the text;

c) The format of the hyphen in Table 1 was inconsistent, e.g. "(14–74)" and "(8 - 80)".

There were some minor grammatical errors.

Author Response

We thank the Reviewer for the time devoted to our research and for his suggestions. In the following we address all the issues raised, highlighting the modification to the manuscript.

1-Firstly, the Introduction section consisted of a single paragraph, which was relatively lengthy and made it easy to lose sight of the main points. The Introduction section of a systematic review article usually has a few catchy paragraphs that are structured and briefly describe the topic or issue surrounding the article

Thank you for your suggestion to make more appealing our work, we have revised the introduction.

2- The Methods section did not provide a clear description of the steps and content of the review, such as the date of the last search, the role of independent reviewers in the selection of articles, the software tools used, and the assessment of the quality and bias of the included articles. In addition, the search of the PubMed database alone was incomplete, with only 311 papers found in the initial search as described in the article.

Thank you for having pointed out these limits, we have improved the methods descriptions. Unfortunately, it has not been possible to proceed to an extensive methodological revision within the limited time allotted to us.

3- Results section: "average extrusion or dislocation rate was 5.1%", "average improvement in ABG for PORP was 12.3 dB", and "average change in ABG for TORP was 13.7", were these average rates or values calculated by performing a meta-analysis? If so, a detailed description of the statistical model, heterogeneity, and software tools would be required in the Methods section, as well as presented as a forest plot in the Results section.

Thank you for the comment, the present topic was approved for the Special Issue “Middle ear and Bone Conduction implants” as a literature review without providing a meta-analysis.

4- Minor: a) The text, such as "(-0,5 dB to 22.7 dB)" and "(CCOM, 34,1%)", appeared in several places, which should be -0.5 and 34.1%; b) The format of references such as "[3], [4]" (Line 41) and "[6], [3]" (Line 179) appeared in two places in the text; c) The format of the hyphen in Table 1 was inconsistent, e.g. "(14–74)" and "(8 - 80)".

Thank you for identifying these inaccuracies, we have corrected it.

5- Comments on the Quality of English Language

We have revised the English

Reviewer 3 Report

This is a review paper on titanium prostheses' dislocation and extrusion following otologic surgery.

The manuscript is written in fine English, but present occasional awkward wording, and should be reviewed by a native English speaker to improve clarity. 

The introduction outlines the rational behind the review and defines the aims and scope of the paper well. The initial section on monaural stimulation is not necessary, since the goal of the review is not central auditory processing. It should rather define the past standard of treatment and compare with current possibilities. 

The paper followed PRISMA guidelines in assebmling the search database, and defined primary and secondary outcomes well. I have no concerns over internal or external manuscript validity. 

The results section defined the review scope well, listing the drop-out ratio, follow-up time, primary and secondary outcome results. When analyzing the table presenting the results, especially regarding dislocation and extrusion, it is worth noticing that many studies reported 0% of events, which is not realistic. This number is almost certainly under-reported, or the follow-up interval was too short. Nonetheless, that could not have been improved by the authors, so it does not represent a bias, but rather a discussion point. 

The discussion then goes on to mention pupillary testing, that is somewhat unrelated to the scope of the review, and does not comment on specific reasons behind extrusion outcomes. This needs to be significantly expanded and points regarding middle ear mechanics, surgical technique and correlations with existing knowledge on middle ear revision surgery and ossiculoplasty need to be specifically discussed. 

Minor wording and phrasing issues. 

Author Response

We thank the Reviewer for the time devoted to our research and for his suggestions. In the following we address all the issues raised, highlighting the modification to the manuscript.

1-The results section defined the review scope well, listing the drop-out ratio, follow-up time, primary and secondary outcome results. When analyzing the table presenting the results, especially regarding dislocation and extrusion, it is worth noticing that many studies reported 0% of events, which is not realistic. This number is almost certainly under-reported, or the follow-up interval was too short.

Thank you for the comment, we have added this point to discussion

2- The discussion then goes on to mention pupillary testing, that is somewhat unrelated to the scope of the review, and does not comment on specific reasons behind extrusion outcomes. This needs to be significantly expanded and points regarding middle ear mechanics, surgical technique and correlations with existing knowledge on middle ear revision surgery and ossiculoplasty need to be specifically discussed.

Thank you for the comment, we have modified the discussion according to your suggestions.

3- Comments on the Quality of English Language Minor wording and phrasing issues.

We have revised the English

Round 2

Reviewer 1 Report

The questions have been correctly answered.

Just a minor correction line 134: a word is missing in the second part of the sentence. 

Reviewer 2 Report

The authors have successfully answered all the questions. 

Reviewer 3 Report

The authors have made the necessary amendments and the improved manuscript is fine. 

Minor revision of language by a native speaker.